# Designing an eHealth Well-Being Program: A Participatory Design Approach

**DOI:** 10.3390/ijerph18147250

**Published:** 2021-07-06

**Authors:** Yannick van Hierden, Timo Dietrich, Sharyn Rundle-Thiele

**Affiliations:** Social Marketing @ Griffith, Griffith Business School, Griffith University, Nathan, QLD 4111, Australia; t.dietrich@griffith.edu.au (T.D.); s.rundle-thiele@griffith.edu.au (S.R.-T.)

**Keywords:** well-being, intervention, eHealth, participatory design, prevention

## Abstract

In recent years, the relevance of eHealth interventions has become increasingly evident. However, a sequential procedural application to cocreating eHealth interventions is currently lacking. This paper demonstrates the implementation of a participatory design (PD) process to inform the design of an eHealth intervention aiming to enhance well-being. PD sessions were conducted with 57 people across four sessions. Within PD sessions participants experienced prototype activities, provided feedback and designed program interventions. A 5-week eHealth well-being intervention focusing on lifestyle, habits, physical activity, and meditation was proposed. The program is suggested to be delivered through online workshops and online community interaction. A five-step PD process emerged; namely, (1) collecting best practices, (2) participatory discovery, (3) initial proof-of-concept, (4) participatory prototyping, and (5) pilot intervention proof-of-concept finalisation. Health professionals, behaviour change practitioners and program planners can adopt this process to ensure end-user cocreation using the five-step process. The five-step PD process may help to create user-friendly programs.

## 1. Introduction

According to the WHO, 93% of countries worldwide disrupted or halted critical mental health services as a result of COVID-19. However, COVID-19 has caused a spike in the demand for mental health services [1] and studies have shown a significant increase in anxiety and psychological distress [2,3,4]. Scholars and experts are asking for new health interventions to support the mental well-being of individuals [5]. One type of intervention that has gained significant popularity is the eHealth intervention. eHealth interventions emerged from the fields of medical informatics, public health, and business, and are characterised as health services and information that are facilitated through the Internet and related technologies [6]. Despite eHealth interventions being around for more than two decades, the literature provides very little evidence of the phases needed to design effective eHealth interventions [7,8], and only a recent review of 46 eHealth interventions [8] identified six phases that are essential when designing an eHealth intervention; namely, design, pretesting, pilot-study, pragmatic trial, evaluation, and post-intervention. However, this review did not mention the importance of user involvement in designing eHealth interventions. Further, many interventions aiming to enhance people’s health and well-being tend to be expert-led in design, with minor or no involvement of its users—potentially limiting their effectiveness. For example, in a review of 77 nutrition and physical activity interventions, only 16% of the interventions conducted some form of formative research with one or more target audiences [9], also demonstrating mixed program effectiveness. Understanding user needs [10,11,12] is essential. However, while application of research to understand users has been identified [13,14], this often does not extend beyond focus groups and surveys [15,16]. For example, in a review of 23 social marketing interventions aiming to minimise harm from alcohol consumption, 57% of interventions used focus groups. While focus groups can provide valuable participant feedback on existing solutions, the participants are not actively involved in intervention design. In a review of 125 health interventions [17], 63% of the reviewed interventions conducted some form of user research; however, research participants were not actively involved in the intervention design processes. Lacking understanding of user requirements and preferences may explain low intervention uptake and retention rates and warrants future research attention [18].

Existing processes that guide the design of health promotion interventions [19,20,21] tend to prescribe expert-led program design methods. Top-down design approaches have been criticised because they do not take into consideration the specific needs and requirements of service users [10,11,12]. For example, mHealth, a pilot study that aimed to measure the effect of a 12-week mobile health intervention, followed an expert-led approach. The study reported no significant results in behaviour change and attributed this failure to the participants’ low engagement with the program [22]. Involvement of the users during the development stage of the intervention may have helped to avoid this costly service failure [23]. Bottom-up approaches that involve users in service design and development processes have emerged and are still gaining popularity [24,25,26]. Babajanian [27] reported that empowerment is an outcome of participation. When bottom-up approaches involve active participation of the community, the results are significantly positive [28]. For example, HEYMAN, a three-month intervention targeted at young men to improve eating habits, activity levels, and well-being [29] demonstrated the feasibility of involving young men in program planning to develop an attractive intervention that assisted young men participating in the program to make positive lifestyle changes.

The importance of user participation is well-evidenced with examples demonstrating that user involvement can engage participants in an intervention [30], increase sense of community [31], increase sense of purpose and commitment [32], strengthen interventions [33], make service delivery more effective [28], and empower people to sustain behaviour over time [34]. The often-used collective noun for collaborative design approaches is cocreation. Cocreation is a collaborative activity in which users actively contribute to the development of a product offering [35]. Design thinking [36], codesign [37], Living Lab [38], human-centred design [39], user-centred design [40] and participatory design [41] all provide methods to ensure that participant voices are heard. These user design approaches embrace diversity and inclusion, ensuring voices are heard, and intervention design is centred on user needs and wants. This paper focuses on participatory design (PD) as one of the most influenced and longest standing cocreation methods.

The term “participation” is frequently used within a broad span of disciplines, including participatory design [42], community-based participatory research [43], participatory action research [44], participatory modelling [45], and participatory journalism [46]. Participation is “a process in which individuals take part in decision-making in the institutions, programs, and environments that affect them” [47]. Participatory design was originally applied in the contexts of human–computer interaction and computer-supported cooperative work in work places [48] but has expanded into areas including healthcare [49], aging and housing [50], product development [51], and infrastructure [52]. Emerging from “design processes” in Scandinavian countries through collaborations between scholars and trade unions, participatory design is now established as a valuable research method with its own methodological orientation, methods, and techniques [41]. Spinuzzi [41] stated that the approach is “as much about design […] as it is about research. In this methodology, design is research”.

PD aims to develop user-centred and effective technologies, products, or services with the active involvement of stakeholders and service users [48]. Given that eHealth interventions require the users to have a good experience, it is imperative to guide eHealth intervention design with insights from human–computer interaction. It is important to note here that a PD process is distinct from the process to develop an eHealth intervention. The six phases suggested by Enam, Torres-Bonilla, and Eriksson [8] can be followed to design an eHealth intervention without the involvement of end-users, whereas a PD process involves users at every phase of the process. A PD process consists of four phases: (1) gathering insights, (2) prototype development, (3) implementation, and (4) evaluation [51,53].

Spinuzzi [41] proposed a three-step process to cover the first two phases of PD; namely, (1) initial exploration of work, (2) discovery processes, and (3) prototyping. However, this process is heavily based on improving existing (computer-based) workplaces and does not translate well to the development of new eHealth interventions. Moreover, this process misses detail in how to gather insights and translate them into prototypes. Sanders, Brandt, and Binder [54] suggested an array of tools and techniques for participatory design approaches; however, when and where to use them remains unclear. A sequential procedural application to cocreating eHealth interventions is currently lacking as a result. According to Sanders and Stappers [37], “everyone has something of value to share at every stage of the process”. Therefore, input from designers, users, and stakeholders should be captured at every stage of the process. As a result, a participatory design process functions as a means to better understand and involve end-users and guide decision making, and hence is an imperative step in creating more tailored and user-friendly products and services [51,55,56]. However, studies tend to involve users in the ideation stage but exclude users from the implementation and evaluation stages of the program/service (see for examples [57,58]). When users are not involved anymore after the service delivery, a wealth of information on the users’ experience is neglected. Experience facilitates empathetic, emotional, and memorable interactions that have intrinsic value [59,60]. Thus, excluding experience from a service design process may limit the creation of effective and user-friendly services. PD recognises that users actively cocreate service experiences [61] and therefore includes participants in all phases from ideation, to design, to development, to implementation, to evaluation.

In conclusion, while four phases of PD are identified (insights, prototype, implement, evaluate), a clear, sequential process to use PD to design eHealth interventions does not exist. Therefore, the objective of this paper is to define a stepwise process to design eHealth interventions using PD. Specifically, this paper provides a five-step process to sequentially design a well-being intervention through the PD phases of gathering insights and proof-of-concept development. The followed PD process leverages the existing three-step PD process [41], yet proposes a more tailored approach to design eHealth interventions to ensure they are cocreated.

## 2. Materials and Methods

A participatory design (PD) research method was employed across five steps to inform the design of a pilot eHealth well-being intervention. The objective of the PD was to identify the most pressing health problems of young adults and involve them in designing new solutions for those problems. This study was conducted in one university in South-East Queensland, Australia. Four PD sessions of ±105 min involving 57 participants aged 18–35 (±14 per session) who experienced prototype activities were conducted in July and September 2019.

### 2.1. Participant Recruitment

Following ethical clearance to conduct the study according to approved protocols, participants were recruited by the research team using lectures, social media, and posters on a university campus. For example, the university Student Association and Student Club Office shared the call for participants on their respective Facebook pages, and A3-format posters were distributed across the university campus. All communication materials had a short URL and a QR code that people could scan to visit a registration page. On this page, the details of the workshop were clearly communicated, and an incentive was offered. Participants were rewarded a $30 voucher in exchange for their participation in this research.

Participation was voluntary, and participants could withdraw from participation without further consequences at any time during the study. A total of 57 participants completed a session. During all sessions, the facilitator emphasised privacy and confidentiality for participants in order to allow them to express their creativity and openly share their thoughts and feelings about sensitive physical and mental health-related materials. The PD participants are a convenience sample. The findings derived from these particular PD sessions and generalisation beyond young adults and other cultural contexts is not possible.

### 2.2. Participatory Design Process

At this point, a distinction should be emphasised between phases and steps. Phases describe a series of events or steps. This paper categorises a PD process into four phases (as described in Figure 1) and each phase can encompass multiple steps. Steps describe specific measures or actions that are taken to achieve an objective. In this paper, a specialised five-step PD process aims to achieve the objective of designing a new well-being intervention. The five-step PD process undertaking in this study covers the first two phases of PD (gathering insights and prototyping) (see Table 1). The latter two phases (implementation and evaluation) are described elsewhere. The applied PD process detailed in the present study built further on the three-step PD process of Spinuzzi [41]. Table 1 contrasts the newly suggested five-step PD process for designing eHealth interventions against the existing three-step PD process of Spinuzzi [41]. In the five-step PD process, we further divide the prototyping phase in multiple smaller steps (steps 3, 4, and 5). Based on the insights from the first two PD sessions (in step 2), a survey was designed to measure psychological well-being [62], mindfulness [63], resilience [64], and physical activity [65]. During the second round of PD sessions (in step 4), participants were challenged to individually design new well-being interventions tailored to their own personal preferences. The results chapter describes the activities of each of the five steps in more detail. The effectiveness of the final proof-of-concept design will be evaluated and reported in a separate study.

## 3. Results

### 3.1. Step 1: Collecting Best Practices

The objective of the eHealth intervention is improved well-being. Therefore, a review of interventions effective in improving well-being was conducted (Table 2).

The review of effective interventions (Table 2) informed the selection of health activities for PD. Then, to select the activities most suitable for the first exploration, three criteria were used. These criteria were (1) evidence-based effectiveness; (2) popularity and broad-based support in society; and (3) experience of the project team.

To meet criterium 1, the effectiveness of health behaviours (from the review in Table 2) were validated by further academic studies (references are provided in the final presentation of activities). Then, to meet criterium 2, popularity of activities was assessed by searching for identified and similar behaviours using Google Trends. To meet criterium 3, one of the members of the research team required some familiarity with the activities to effectively facilitate the participatory design sessions (i.e., guiding participants through an experience of the selected health behaviours). Based on these criteria, the following activities were selected: relaxation exercises [75], physical activity [76], meditation [77], sleep habits [78], and actions intended to benefit others (acts of kindness) [79]. These ideas were then brought forward into step 2.

Previous research has shown that theory-based interventions tend to have larger effects than interventions that are not developed under the guidance of theory [80,81,82]. However, it is concerning that interventions rarely use theory (for reviews, see [81,83]). For example, a meta-analysis by Prestwich, Sniehotta, Whittington, Dombrowski, Rogers, and Michie [84] demonstrated that only fifty-six percent of health behaviour interventions reported a theory base. This finding warrants the inclusion of a behaviour change theory in the design of the eHealth well-being intervention.

### 3.2. Step 2: Participatory Design Sessions 1 and 2—July 2019

Results from the four PD sessions involving 57 participants (60% female) are presented in sequence of the five PD steps (see Table 1). During the two sessions in step 2, participants were guided through a breathing exercise of ±7 min. Then, participants were guided through a sequence of physical activities (i.e., push-ups, planking, and yoga poses) for ±7 min. Then, participants were shown a video of ±7 min during which the researcher explained a concept related to psychological well-being and addressed the importance of lifestyle habit change.

In session 1, the concept was digital minimalism, and the video explained that new technology—especially smartphones—have impacted our (mental) health and how digital minimalism is an approach to overcome that negative impact by creating a healthier relationship with technology [85]. Following the viewing of the video, the facilitator explained how participants could adopt the discipline of digital minimalism by partaking in the “digital declutter”. The digital declutter is a 30-day challenge that asks its participants to follow three steps: (1) take a break from optional tech—new technologies that participants can take a break from without doing major harm to their social or professional lives (e.g., deleting the Facebook or Instagram application from their phone for 30 days); (2) actively engage in real-life activities, such as physical activity, outdoor activity, and spending time with friends, family, and community (e.g., woodworking, planting trees, voluntary service, etc.), and identify which activities are meaningful and derive deep value; and lastly, (3) after 30 days, reintroduce only the new technology that contributes to the identified deep values and develop a procedure that determines when and how to use it.

In session 2, the concept was service, and the video explained that “we can each affect our happiness and the happiness of those around us” and “helping others is essential for a healthier and happier society.” [86]. Following the viewing of the video, the facilitator explained how participants could start doing things for others by implementing acts of kindness—small acts that people can do for others around them (e.g., finding out about the values of another culture, doing something to help a project or charity you care about, etc.)—and keeping a daily journal of these activities.

In the second step of the sessions, after experiencing the above-described activities, participants were asked to form groups of three people. Individuals were asked to fill in a feedback capture grid [87]—this grid is divided into four quadrants (likes, criticism, questions, and ideas). Then, participants were asked to form groups and were challenged to give the experienced exercises names and state their willingness to pay for and willingness to spend time on the program. Finally, they were asked whether they would rather engage with the program face-to-face or online and for how long. In the third and last step of the sessions, participants were asked to design a promotion campaign. Table 3 provides a summary of the conducted activities to easily distinguish session 1 from 2.

During the first two PD sessions with 27 participants, people smiled and laughed during the icebreaker activities and thus seemingly enjoyed themselves. During the experiences of the potential intervention activities, participants were all actively involved and intrigued by the experiences (see Figure 2). During the individual feedback sessions, they seemed serious and concentrated in giving feedback. During the group work sessions, participants were highly engaged and actively interacted with each other. These observations were confirmed by the feedback people gave. When asked what people liked, one participant said, “loved the meditation–made me feel very relaxed”. Another commented, “[the information on digital minimalism] provided deep insights into how digital media has impacted our inner happiness”. Another remarked that they were “able to calm down and be centred”. When people were asked what they disliked, one participant said, “it needs to be more fun”. Another commented that they wanted “to have more time”. Another remarked that they “wanted to have more activities”. Participants also proposed interesting ideas; namely, to include a social component, to start the intervention with a self-assessment, and to have a means of tracking progress throughout the program. Willingness to pay varied significantly. In the first session, participants were willing to pay between $10–15 per module/week or between $40–60 for the entire program. In the second session, participants suggested different intervention formats. For example, one participant would pay $7.99 per month if it were a mobile app. Another participant would pay $30 per fortnight, $50 per month, and $500 per year if it were an ongoing service. One person was not willing to spend any money, and another was willing to spend $2000 on the entire program. On average, people were willing to invest 20 min per day on the program. Some people thought an ongoing program was desirable, while others thought a program length of four weeks was sufficient. The most preferred program components were deep-breathing exercises, martial arts and yoga, lifestyle habits, digital minimalism, meditation, sleep habits, and performing acts of kindness.

### 3.3. Step 3: First Design of Intervention Proof-of-Concept

The data collected in step 2 was analysed using thematic coding analysis. Using NVivo Version 12, the data was coded, which allowed the researchers to identify and prioritise themes that emerged from the data [88]. By addressing the participants’ preferences, including their questions and ideas, a structure of activities emerged, and a first overview of preferred delivery, duration, and price became apparent. These data were used to generate a full 4-week intervention proof-of-concept for the next PD phase (see Figure 3).

The findings from step 2 were used to generate a full 4-week intervention initial proof-of-concept. Based on the likes, dislikes, criticism, and ideas from participants, a number of activities were combined into a face-to-face intervention delivery. The intervention would consist of four workshops lasting 60–90 min each. It would always start with a deep-breathing exercise, then go into physical activity (e.g., tai chi, yoga asanas, kung fu, and qigong). Then, the workshop would transition to a lecture format and would provide people with lessons on digital minimalism, mastering health, mindfulness, and service. The workshops would all finish with an activity-based component to build new habits around the topic of the workshop (e.g., 30-day digital declutter, habits around exercises, nutrition and sleep, habits to master your mind, and habits to perform acts of kindness).

### 3.4. Step 4: Participatory Design Sessions 3 and 4—September 2019

Where step 2 focused on the participant’s experience of and feedback and ideas on activities, step 4 focused on finalising the proof-of-concept design of the overall intervention in terms of structure, length, delivery, and price. During the sessions in step 4, participants were first asked to complete a questionnaire. After completing the questionnaire, the participants were guided through a 7–10 min breathing exercise similar to those in step 2. Following the breathing exercise, the facilitator presented multiple activities that could be part of the intervention. These activities included the most popular activities identified in step 2 and additional suggestions given in those sessions. For an overview of the PD session activities, please refer to Table 3. Then, participants were asked to individually complete the feedback capture grid. After completing the feedback capture grid, the participants were asked to individually design their own programs (see Figure 4). In this step, 27 unique programs were designed (see Figure 5 for an example).

During the last two PD sessions with 30 people, participants went through similar experiences as during the first sessions (Table 4). When asked what people liked, some participants mentioned “face-to-face delivery”, and others mentioned “the combination of mental and physical health”, while another person mentioned “the essentials of fulfilling purpose and meaning”. Some people liked the ability to track their progress, to self-reflect, and to perform acts of kindness. When asked what people disliked, some people commented that one workshop per week would be too short to go through all the activities effectively. They opted for more workshops or workshops of longer duration. Engaging with a community post intervention online was viewed favourably. The intervention implementation length would be 5 weeks, extending the initial proof-of-concept length by one week.

### 3.5. Step 5: Final Draft of Intervention Proof-of-Concept

In step 5, data was content analysed [89] using SPSS Version 25 (IBM Corp: Armonk, NY) and NVivo Version 12 (QSR International: Melbourne, Australia). This analysis identified most preferred proof-of-concept activities, leading to an outcome proof-of-concept (see Figure 6), which will be used to guide intervention build for the implementation phase of the PD process.

The findings from the previous steps were used to alter the proof-of-concept to a 5-week eHealth well-being intervention. The final components of the intervention would be delivered through an eLearning portal. Accommodating the needs of the participants, each week of the program would entail 90–120 min of digital content (video lessons, quizzes, and worksheets) to deliver the program components. A fifth week was added, which would focus on reviewing progress and preparing to keep consistency moving forward. More specifically, the 30-day digital declutter would be reduced to 7 days; participants would receive a personal journal; and attention would be focused on setting goals, building habits (e.g., exercise, sleep, mindfulness, meditation, voluntary service, and acts of kindness), reviewing progress, and keeping consistency. Other aspects such as a self-assessment, socialising, professional assistance, and lifestyle design, would be included. In addition, a virtual community would be developed to enable online community engagement and provide further resources to help people to continue to improve their well-being. A visual design of the final proof-of-concept can be seen in Figure 6.

## 4. Discussion

A sequential procedural application to cocreating new eHealth interventions is currently lacking. Existing PD processes lack detail in terms of operationalisation and do not transfer well to the development of new eHealth interventions [41]. This paper demonstrates implementation of a more detailed and sequential participatory design (PD) process to inform intervention design. This paper advances understanding of how to design new eHealth interventions using participatory design by delivering two contributions. First, this paper details a five-step process to sequentially apply participatory design to the design of a new eHealth well-being intervention. Second, the five-step PD process may help to create user-friendly programs. These contributions are described in more detail hereafter.

First, this paper provides a five-step process that can be sequentially applied ensuring that a participatory design approach is applied to design new eHealth well-being interventions (Figure 7). The five step PD process is (1) collecting best practices (2) participatory discovery; (3) initial proof-of-concept; (4) participatory prototyping; and, (5) pilot intervention proof-of-concept finalisation. This new process differs from existing methods in [41] in several ways. First, this PD process enables the collection of insights extending application beyond examining existing workplaces. Interventions that are designed from scratch, and intervention designers without access to existing workplaces, need processes that guide the collection of insights that can draw on the strength of existing research and best practices to understand what works, when, and why. Moreover, this process demonstrates how insights can be collected by actively involving users in the experience of intervention activities, generating valuable insights that can be likened to an existing workplace. Therefore, this study provides guidelines and tools for experiential user involvement. This study demonstrates that when users experience activities, their creative capacity to cocreate value is increased. This is in line with work by Lusch and Vargo [60] as well as Ranjan and Read [90].

Second, health professionals, behaviour change practitioners, and intervention planners can adopt the five-step process described in this study to develop proof-of-concept programs, delivering a template that can be used by teams to design e-health interventions. This process may help to create user-centred interventions. Involving people in PD activities may enable them to form accurate judgements of the suitability, likeability, feasibility, and effectiveness of potential service components. Furthermore, the researchers took notes and video recorded the majority of the sessions, which enabled careful observation of the participants while they were engaging in experiences, providing researchers with data that could not be captured in written or spoken feedback by participants. Expressions and body language of participants after they completed exercises enabled researchers to determine whether participants liked or disliked potential components. Therefore, participatory design methods, in which users experience activities, extend insights enhancing intelligence for eHealth well-being intervention development.

## 5. Conclusions

This study demonstrates how PD tools can be used to gain user insights to guide proof-of-concept development. A stepped process is described demonstrating how teams can transition from insights to iterated proof-of-concepts. This process enables intervention designers to address user needs and to build proof-of-concepts reflecting user preferences, ensuring eHealth interventions are cocreated [5]. Practitioners who are designing an eHealth well-being intervention for young adults are advised to ascertain whether the proposed service components described in this study are of use for them. Additionally, they should carefully gauge the balance of online and offline content delivery to ensure that user preferences and cost effectiveness in intervention creation are maintained. They should include experiential activities to facilitate learning and include options to assess well-being measures and review progress throughout the intervention. Furthermore, it is imperative that ongoing support is offered in the form of a community or an online platform with tools and content.

As with all studies, this study is not without limitations. The findings were derived from these particular PD sessions, and generalisation beyond young adults and other cultural contexts is not possible. Furthermore, the researcher selected some but not all identified activities from the literature review to be included in the PD sessions. While the specific activities were deemed popular and effective by participants, they might not have fulfilled all participant preferences. eHealth well-being interventions guided by PD approaches may be more effective in changing well-being behaviour. However, further research is required to support this claim. Such research should evaluate the effectiveness of interventions that followed a PD approach and interventions that followed an expert-led approach. Then, it would be possible to compare effectiveness in terms of improved well-being and user uptake and retention. Furthermore, it is recommended to pilot test the intervention proof-of-concept to assess the uptake, retention, and effectiveness of the intervention. Generalisability of findings in this study could be tested further through preference elicitation methods. For example, preferred activities and recommendations could be examined using Best Worst study or conjoint analysis methodologies in larger user samples prior to intervention development.

## Figures and Tables

**Figure 1 ijerph-18-07250-f001:**
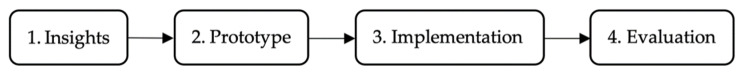
Four phases of a participatory design process (identified from [52,54]).

**Figure 2 ijerph-18-07250-f002:**
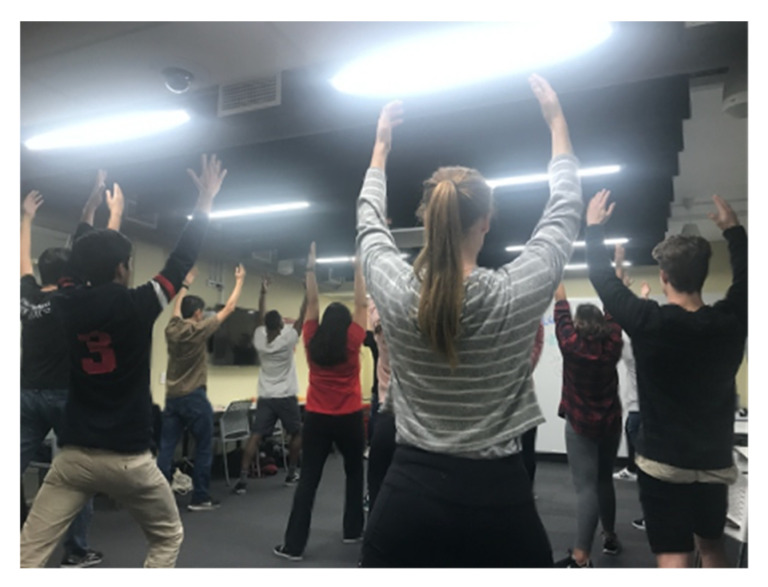
PD session participants doing yoga poses.

**Figure 3 ijerph-18-07250-f003:**
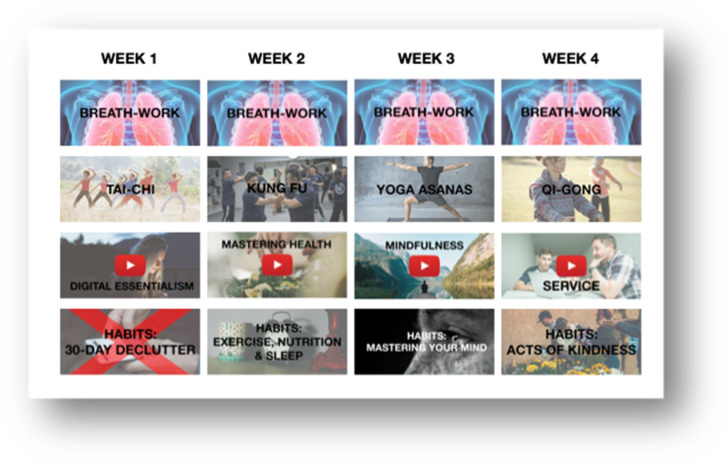
First proof-of-concept.

**Figure 4 ijerph-18-07250-f004:**
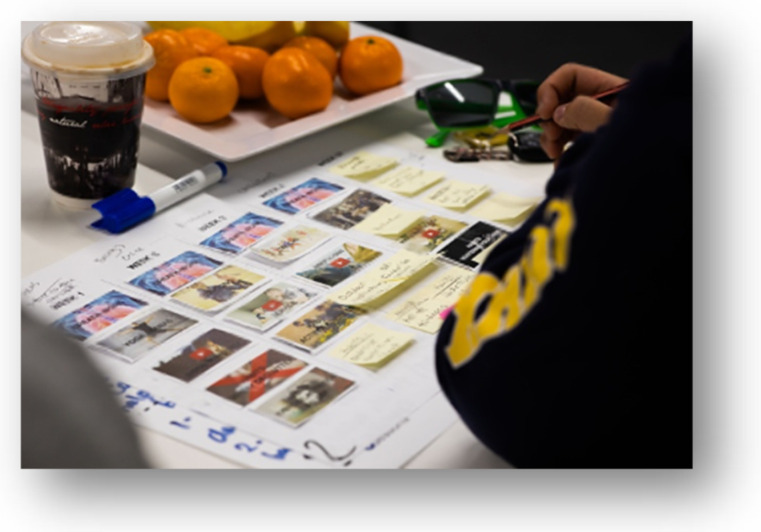
PD participant designing a well-being program.

**Figure 5 ijerph-18-07250-f005:**
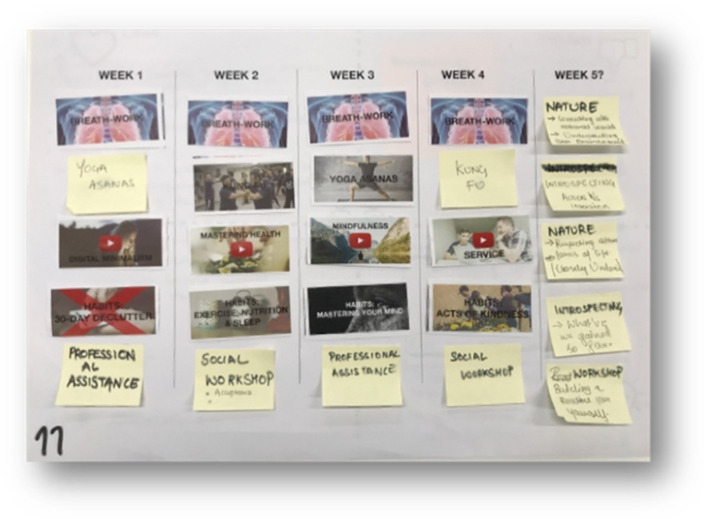
Example of a participant program design.

**Figure 6 ijerph-18-07250-f006:**
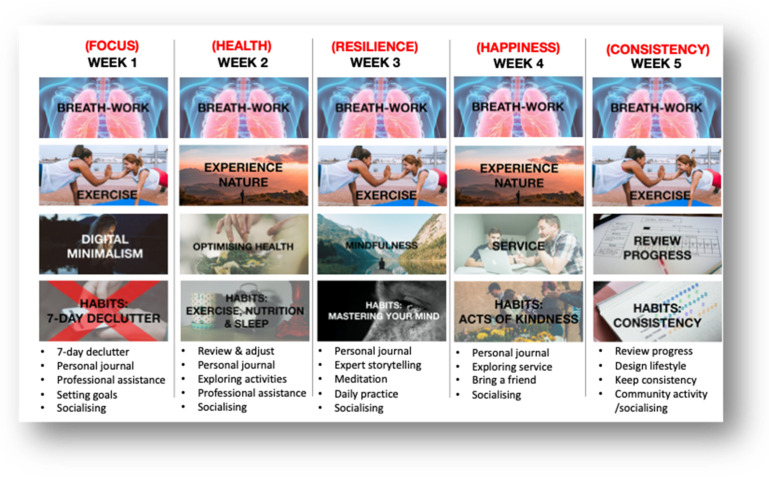
Final program prototype.

**Figure 7 ijerph-18-07250-f007:**
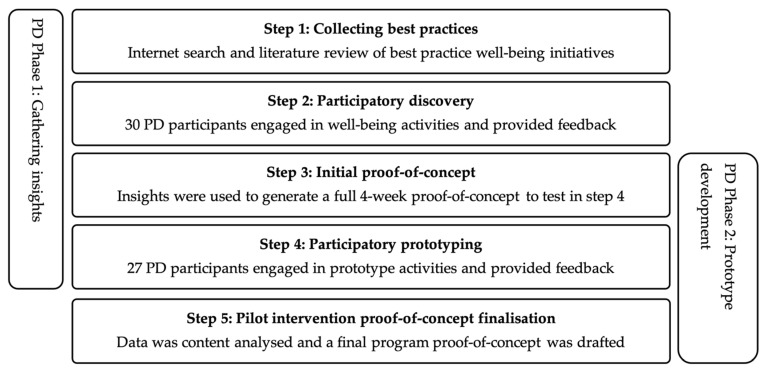
Overview of the five-step PD process to design eHealth interventions.

**Table 1 ijerph-18-07250-t001:** Distinction Spinuzzi’s three-step PD process and the five-step PD process for eHealth interventions.

Overall PD Phases (Figure 1)	Spinuzzi’s Three-Step PD Process	Five-Step PD Process for eHealth Interventions	Rationale for a New PD Step
Phase 1: Gathering insights	1. *Initial exploration*: explore workplace to assess current work processes.	1. *Collecting best practices*: exploring best practice well-being activities through secondary research.	The objective of this study was to design a new eHealth intervention. Hence, insights could be collected beyond existing workplaces. This step was necessary to inform the second step.
	2. *Discovery process:* review existing work processes and envision a future workplace.	2. *Participatory discovery*: elicit people’s preferences regarding well-being best practices and gain ideas for the design of a new intervention.	Given that the focus was on designing a new intervention, PD sessions (with 30 participants) could identify people’s preferences regarding effective well-being activities.
Phase 2: Prototype development	3. *Prototyping:* iteratively shaping artefacts	3. *Initial proof-of-concept*: using people’s preferences to inform an initial proof-of-concept for PD sessions	Because we had utilised the first PD sessions as a means to elicit preferences and ideas for a prototype, we were able to use that information to suggest an initial proof-of-concept for the second round of PD sessions.
	–	4. *Participatory prototyping*: collecting feedback on proof-of-concept and the cocreation of new intervention designs according to people’s personal preferences	This extra step with two PD sessions was deemed necessary to ensure the emergence of a cocreated, user-friendly proof-of-concept.
	–	5. *Pilot intervention proof-of-concept finalisation*: finalise an outcome proof-of-concept based on PD insights.	It is imperative to inform the intervention design on the basis of user preferences and their ideas; however, program designers must carefully gauge available resources (e.g., time and budget) in the development of the proof-of-concept that prevent the uptake of some ideas proposed by users. Therefore, an extra step was added to finalise the outcome proof-of-concept.

**Table 2 ijerph-18-07250-t002:** Review of effective well-being interventions.

Campaign/Year/ Author/Organisation	Targeted Health Behaviour	Outcome/Evaluation
The Student Compass [66]	Journaling, relaxation exercises	After a 7-week intervention, participants showed significantly higher gains in well-being, life satisfaction, and mindfulness skills. In addition, iACT participants’ self-reported stress and symptoms of depression were significantly reduced
Gratitude Group Program [67]	Gratitude journaling	After a 5-week intervention, participants showed a significant and clinically meaningful decrease in psychological distress and increase in state gratitude, satisfaction with life, and meaning in life
Happiness 101 [68]	Mindfulness, gratitude, goal setting	After a 6-week intervention, scores improved from baseline to 6-month follow-up for health, vitality, mental health, and the effects of mental and physical health on daily activities. Improvements in mental and physical health and functioning were shown over a 6-month period
HEYMAN [29]	Nutrition, physical activity	After a 3-month intervention, significant effects were found for daily improving vegetable servings; energy-dense, nutrient-poor foods; weight; BMI; fat mass; waist circumference; and cholesterol
Internet-based mindfulness training program [69]	Mindfulness	Both the basic and HAPA-enhanced mindfulness groups showed better mental well-being from pre-intervention to post-intervention, and improvement was sustained at 3-month follow-up
Online behavioural weight management program for college students [70]	Nutrition, Physical activity	Overweight/obese students lost an average of 5.1 ± 6.0 lbs. Those of healthy weight lost an average of 1.8 ± 3.2 lbs. Twenty-three percent of students lost >5% of their baseline weight
Print- and Internet-Based Physical Activity (PA) Promotion Intervention [71]	Physical activity	At 6 months, the tailored internet arm reported 120 min of PA/week, and the tailored print arm 112.5 min of PA/week. At 12 months, the physical activity minutes per week were 90 for both interventions
An Electronic Wellness Program to Improve Diet and Exercise in College Students [72]	Nutrition, physical activity	Mean change from baseline of saturated fat intake was marginally significant between the treatment groups at week 24. A significant difference in percent of snacks chosen that were fruit was detected
RCT of a Smartphone-Based Mindfulness Intervention [73]	Mindfulness	Positive affect with a medium effect size and reduced depressive symptoms with a small effect size
The Well-being Game [74]	Journaling, physical activity	Students reported a significant positive change in well-being levels; employees reported lower stress levels and higher well-being levels

**Table 3 ijerph-18-07250-t003:** Outline of sessions in step 2.

	Session 1	Session 2
Part 1: Participation	Experiencing activities	Experiencing activities
Breathing exercises	7 min breathing exercise	7 min breathing exercise
Physical activity	10 min physical activity routine	10 min physical activity routine
Video viewing	Digital minimalism	Service
Action habits	Digital declutter	Acts of kindness, journaling
Part 2: Feedback capture	Feedback Capture Grid	Feedback Capture Grid
Part 3: Promotion campaign design	Campaign design	Campaign design

**Table 4 ijerph-18-07250-t004:** Outline of sessions in step 4.

	Session 3 and 4
Part 1: Questionnaire	10 min questionnaire related to psychological well-being, mindfulness and resilience, and physical activity
Part 2: Breathing exercise	7–10 min breathing exercise
Part 3: Feedback capture grid	Feedback capture grid based on presented design
Part 4: Program design	Individual design of most preferred program

## Data Availability

The data presented in this study are available on request from the corresponding author.

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
