# Peer review of "Designing an eHealth Well-Being Program: A Participatory Design Approach"

_ijerph, 2021, doi:10.3390/ijerph18147250_

Round 1

Reviewer 1 Report

The paper describes the implementation of a new type of participatory design for an eHealth intervention.

I think that the paper needs some improvements before to be considered for acceptance.

First of all, I have noticed in Section 1 that, while the characteristics of PD and the gaps in the literature are well discussed, the concept of eHealth intervention is overlooked. The section also lacks in explaining the features an eHealth intervention should have. In particular the paper remains unclear how the Authors conceptualized their eHealth intervention in relation to the implemented PD. I suggest that the Authors improve Section 1 adding a paragraph explaining the concept of an eHealth intervention and the characteristics that an eHealth intervention should have in the light of the literature. I think that this improvement will be very helpful in order to understand the appropriateness of the stages that the Authors have conceptualized for their PD.

Moreover, I think that the paragraph in l. 118-124 is unclear and repetitive. The Author did not univocally define the objectives of their study. I suggest that the Authors improve the last paragraph of Section 1, or better that they clearly stated the objectives of their study. It is still not clear to me if the objective of the paper is the design of a new eHealth intervention (in this case the goals of the eHealth intervention should be defined), or the elaboration of a new and improved PD. However, in this latter case, the Authors proposes a single PD, without specifying how they defined the proposed PD and its objectives, and how they can evaluate it in comparison with other different PDs.

Regarding Section 2, I was wondering how the Authors defined the 5 stages and why they decided that the PD consists of 5 stages. I think that the Authors should discuss from a theoretical and practical point of view why they proposed that PD and how they framed the structure of their PD.

The process represented in Table 1, and explained at pages 3 and 4, is proposed as granted. But, if previous studies are lacking, the Authors should explain why they decided this type of PD. I recommend that the Authors also better explain the motivation behind their decisions.

The Authors should specify that the sample is a convenience sample, and that the validity of their experiment can be reduced by the participants recruitment they adopted. This should be included also in Section 2 in my opinion and not only at the end of the paper.

I was wondering if the Authors have collected information about participants’ health needs and lifestyles. These aspects can influence their preferences of the proposed interventions.

Section 3 reports the 5 stages of the PD. Considering how the Authors shaped this section, I confirm my suggestion that the Authors should explain how they reached the decision on how to organize the PD and which activities to be included in the programs.

Lines 180 – 188: the Authors should better detail their review of the literature and how they came to find the three criteria from the literature. They should also explain how they selected the activities and their link with the three criteria.

Lines 261-263: the Authors reported here the same citations that they already included at page 5.

Finally, I think that Section 4 should be revised by the Authors. They described the advantages of their PD and eHealth intervention. However, they adopted a single PD and no alternative PDs are implemented in this study. Therefore, comparisons (see for example at line 371 “more user-friendly programs” or line 388 “more accurate judgement”) or improvements (see for example at line 383 “can be enhanced by actively involving users…”) can not be confirmed.

Reviewer 2 Report

This study described a comprehensive PD process of an online intervention program. This has allowed the designers to better integrate user needs in the intervention.

A minor suggestion is to include more literature on “user-centered design” and “HCI” and expand the background section a bit more. One to two paragraphs would be great to add another theoretical layer to the PD of an online/eHealth program.

Round 2

Reviewer 1 Report

Thank you very much for the manuscript revision. I appreciated changes and improvements. 

I have just a few minor modifications:

  • I suggest that the authors move lines 197-198 after the subsection 3.1 to subsection 3.2. They should check if the indication Figure 2 on p. 7 is correct (line 198).
  • servies line 96
  • figure 1 is cited twice at line 170

Author Response

Thank you very much for your positive response and the minor modifications. We had adopted your suggestions and made the following changes:

We have now placed lines 197-198 to 225-22: Results from the four PD sessions involving 57 participants (60% female) are presented in sequence of the five PD steps (see Table 1) 

Thank you for picking up the Figure reference. We have updated the reference from Fig. 2 to Table 1 (line 225-226)

We have corrected line 96 to 'services'

We have removed the double Figure 1 at line 170.

Thanks again for your helpful comments, we very much appreciate your time and thoroughness in review.